# Systematic Analysis of Genetic and Pathway Determinants of Eribulin Sensitivity across 100 Human Cancer Cell Lines from the Cancer Cell Line Encyclopedia (CCLE)

**DOI:** 10.3390/cancers14184532

**Published:** 2022-09-19

**Authors:** Pallavi Sachdev, Roy Ronen, Janusz Dutkowski, Bruce A. Littlefield

**Affiliations:** 1Eisai Inc., Nutley, NJ 07110, USA; 2Data4Cure Inc., La Jolla, CA 92037, USA; 3Eisai Inc., 35 Cambridgepark Drive, Cambridge, MA 02140, USA

**Keywords:** eribulin, paclitaxel, vinorelbine, gene expression, predictive biomarkers

## Abstract

**Simple Summary:**

Despite decades of clinical use and detailed understandings of their mechanisms of action, clinically useful predictive biomarkers for approved microtubule targeting agents such as eribulin, paclitaxel and vinorelbine have eluded development. Our results now provide the basis for gene expression-based, drug-specific predictive biomarkers for eribulin and vinorelbine, as well as deeper understandings of the molecular pathways associated with eribulin and vinorelbine response.

**Abstract:**

Eribulin, a natural product-based microtubule targeting agent with cytotoxic and noncytotoxic mechanisms, is FDA approved for certain patients with advanced breast cancer and liposarcoma. To investigate the feasibility of developing drug-specific predictive biomarkers, we quantified antiproliferative activities of eribulin versus paclitaxel and vinorelbine against 100 human cancer cell lines from the Cancer Cell Line Encyclopedia, and correlated results with publicly available databases to identify genes and pathways associated with eribulin response, either uniquely or shared with paclitaxel or vinorelbine. Mean expression ratios of 11,985 genes between the most and least sensitive cell line quartiles were sorted by *p*-values and drug overlaps, yielding 52, 29 and 80 genes uniquely associated with eribulin, paclitaxel and vinorelbine, respectively. Further restriction to minimum 2-fold ratios followed by reintroducing data from the middle two quartiles identified 9 and 13 drug-specific unique fingerprint genes for eribulin and vinorelbine, respectively; surprisingly, no gene met all criteria for paclitaxel. Interactome and Reactome pathway analyses showed that unique fingerprint genes of both drugs were primarily associated with cellular signaling, not microtubule-related pathways, although considerable differences existed in individual pathways identified. Finally, four-gene (*C5ORF38*, *DAAM1*, *IRX2*, *CD70*) and five-gene (*EPHA2*, *NGEF*, *SEPTIN10*, *TRIP10*, *VSIG10*) multivariate regression models for eribulin and vinorelbine showed high statistical correlation with drug-specific responses across the 100 cell lines and accurately calculated predicted mean IC50s for the most and least sensitive cell line quartiles as surrogates for responders and nonresponders, respectively. Collectively, these results provide a foundation for developing drug-specific predictive biomarkers for eribulin and vinorelbine.

## 1. Introduction

Microtubule targeting agents (MTAs), including eribulin, several taxanes, vinca alkaloids, and at least one epothilone, are important clinical anticancer drugs for several cancer types [1,2,3,4]. Despite sharing targets of MTs and their α/β-tubulin building blocks, different MTAs have different clinical profiles, binding sites, mechanisms of inhibiting MT dynamics, as well as different effects on mitotic versus interphase cells [1,2,3,4]. For example, paclitaxel is an MT stabilizer, while eribulin and vinorelbine are MT destabilizers; both MT stabilization and destabilization are sufficient to disrupt the MT dynamics that underly MT function, yet their mechanisms of doing so are quite different [1,2,3,4]. While such differences must certainly contribute to the different clinical profiles seen with these agents, to date, knowledge of differences at the molecular, biochemical and cellular levels has not yielded reliable and clinically useful predictive biomarkers for any MTA. Thus, despite decades of clinical use, the need for predictive biomarkers to identify patients most likely to respond to specific MTAs remains.

Eribulin, a synthetic analog of the marine sponge natural product halichondrin B [5], is an MT dynamics inhibitor with both cytotoxic antimitotic mechanisms [6,7] and noncytotoxic effects on tumor vasculature, tumor phenotype and the tumor immune landscape [8,9,10,11,12,13]. Eribulin’s unusual combination of cytotoxic and noncytotoxic effects suggested that identifying determinants of eribulin response might be possible given a sufficiently broad survey of cancer cell lines with differing sensitivities and baseline gene expression patterns. Accordingly, we quantified in vitro antiproliferative responses to eribulin, versus paclitaxel and vinorelbine as clinically relevant comparators, in 100 human cancer cell lines from the Cancer Cell Line Encyclopedia (CCLE) [14] and correlated the results against the expression of 11,985 genes in the publicly available DepMap database to assess genetic and pathway determinants of eribulin response. Here, we report the identification of small sets of drug-specific unique fingerprint genes (UFGs) that strongly correlate with drug-specific responses to eribulin and vinorelbine. Network propagation and Reactome pathway analyses seeded by these UGFs pointed to cellular signaling, not MTs and mitosis, as the most dominant overall theme driving response to both drugs. Finally, multivariate regression (MVR) models constructed from four- and five-gene UFG subsets correlated with drug-specific responses across all 100 cell lines and accurately predicted mean response levels (IC50s) of the most and least sensitive cell line quartiles as modeling surrogates for responder and nonresponder populations.

## 2. Materials and Methods

### 2.1. Test Agents

Eribulin mesylate (hereafter, eribulin) was supplied by Eisai Inc. (Cambridge, MA, USA) as laboratory-grade dry powder active pharmaceutical ingredient (API) obtained from the same manufacturing stream used for Eisai’s branded clinical product, Halaven^®^. Paclitaxel and vinorelbine tartrate (hereafter, vinorelbine) were obtained as dry powders from Selleckchem (Shanghai, China). Eribulin, paclitaxel and vinorelbine were prepared as 10 mM stock solutions in 100% (*v*/*v*) DMSO, aliquoted into small volumes and stored at −20 °C until day of use. Although not a designated test agent for this study, cisplatin was included in all assays as an internal reference control for assay performance; cisplatin was obtained as a laboratory grade liquid formulation from Hospira Australia Pty Ltd. (Melbourne, Australia), with storage per manufacturer’s instructions and dilutions for cell culture studies on day of use.

### 2.2. Human Cancer Cell Lines

One hundred established human cancer cell lines were selected based on inclusion in both the CCLE and Crown Bioscience’s OmniPanel^TM^ in vitro cell line screening service (see Section 2.3). A list of selected cell lines and their tissues of origin are presented in Appendix A. Cell line selection was based on a combination of common usage, well-established characterization in the literature, personal experience and our hypothesis that the most diverse cell line panel would provide the highest likelihood of identifying genes and pathways associated with specific drug responses, agnostic of cancer type.

### 2.3. In Vitro Cell-Based Antiproliferative Assays

Measurement of antiproliferative activities of test agents against the selected 100 CCLE cell lines was performed by Crown Bioscience under contract from Eisai Inc., using Crown’s OmniPanel^TM^ in vitro cell line screening service in their Beijing (China) laboratories. All OmniPanel^TM^ cell lines are routinely tested for mycoplasma and authenticated using short tandem repeat (STR) DNA profiling. Testing was conducted using the CellTiter-Glo^®^ Luminescent Cell Viability Assay (Promega Corp., Beijing, China) following 72 h compound exposures in 96-well plate assay formats. Compounds were added 24 h after cell seeding into plates, with initial seeding densities for individual cell lines having been previously optimized for OmniPanel^TM^ screening. Based on the authors’ prior experience with the test agents, assays employed 9-step half-log test concentrations of 30 pM–300 nM for eribulin and 100 pM–1 μM for paclitaxel and vinorelbine. Concentrations of test agents inhibiting 50% of viable cell densities in vehicle-treated control wells compared to wells with the highest drug concentrations were defined as IC50s and were determined using GraphPad Prism software (version 5.0; GraphPad-Prism China, Beijing, China). Dose–response curve fitting used a nonlinear regression model with sigmoidal dose response, fixed 100% top *y*-axis values defined by vehicle and floating bottom *y*-axis values at either the highest concentration tested or another concentration that resulted in a minimum *y*-axis reading, ensuring that calculated IC50s represent actual antiproliferative biological responses occurring during the 72 h treatment period without plateau artifacts based on different growth rates of different cell lines.

IC50s for 5 cell lines for eribulin (769-P, 786-O, CADO-ES1, HCT-15, NCI-H716) and 1 cell line for paclitaxel (HCT-15) exceeded the 300 nM and 1000 nM top concentrations used for these 2 drugs, respectively, so surrogate values of 301 nM eribulin and 1001 nM paclitaxel were assigned for purposes of defining cell lines in the bottom (least sensitive) quartile and calculating means/medians of bottom quartiles. Although using such surrogates does not change assignment of bottom quartiles or median IC50 values, it probably results in minor underestimations of mean bottom quartile IC50s, SDs and SEMs for eribulin and paclitaxel, although such underestimations are likely to be small due to the large overall number of cell lines tested.

### 2.4. Gene Expression Data Analysis

Expression data for 11,985 genes from the 100 selected cell lines were downloaded from the Cancer Dependency Map (DepMap) project portal (https://depmap.org/portal/ accessed on 3 August 2021; release version “DepMap Public 18Q2”, 2 May 2018). DepMap data using the symbol *SEPT10* were replaced with the currently accepted symbol *SEPTIN10*. Expression values are log_2_ gene-level reads per kilobase million (RPKM) derived from RNA Sequencing (RNA-Seq), aligned using TopHat version 1.4 and quantified using the pipeline developed for the Genotype-Tissue Expression (GTEx) project as described by Tsherniak et al. [15]. Systems-level analyses of genes associated with drug response were performed using the Data4Cure Biomedical Intelligence^®^ Cloud (La Jolla, CA, USA) [16] to identify gene-level determinants of response that were unique to eribulin or shared with vinorelbine or paclitaxel (see Section 2.5). Differential gene expression analysis comparing top and bottom cell line quartiles for each drug (most and least sensitive 25 cells lines, respectively) was performed using the limma R package [17], fitting a linear model for each gene and producing a moderated t-statistic, fold-change, *p*-value and q-value for each gene, with q-values representing *p*-values adjusted for multiple hypothesis testing using the false discovery rate (FDR) method of Benjamini and Hochberg [18].

### 2.5. Identification of Drug-Specific UFGs

Ratios of mean expression levels for each gene between the most and least sensitive cell line quartiles (top and bottom quartiles, respectively) for each drug were calculated. Genes with top/bottom quartile ratios with *p* < 0.0025 significance for a given “cognate” drug were further restricted by sequentially (i) excluding genes having *p* < 0.0025 overlaps with either of the 2 “noncognate” drugs, (ii) retaining only genes with either ≥2× increased or ≤0.5× decreased expression ratios between top and bottom quartiles (|log_2_[quartile expression ratio]| ≥ 1) and (iii) after reintroducing data from the middle two cell line quartiles, retaining only those genes with statistical significance (*p* < 0.05) for the cognate drug in full linear regression analysis with all 100 cell lines, but lacking statistical significance for both noncognate drugs. Genes meeting all the above criteria for a given drug were designated as unique fingerprint genes or UFGs for that drug. No gene met all criteria for paclitaxel; thus, no paclitaxel UFGs were identified in this study.

### 2.6. Network Propagation and Reactome Pathway Analyses

Network propagation [19] starting with the 9 eribulin UFGs and 13 vinorelbine UFGs as query inputs was performed using the Network Enrichment platform in the Data4Cure Biomedical Intelligence^®^ Cloud [16]. The resulting 100-gene networks were then used as query inputs for Reactome pathway analyses [20,21] using the Reactome portal in the Data4Cure platform.

### 2.7. Multigene MVR Model Building

MVR model building was performed to develop multigene panels of UFGs that can predict the likelihood of high versus low response to eribulin and vinorelbine (MVR model building was not performed for paclitaxel since no gene met all UFG criteria for that drug). Full sets and subsets of eribulin and vinorelbine UFGs were combined as follows. Since the criteria for UFGs demand that expression levels of each UFG show statistically significant correlation with IC50s in linear regressions across all 100 cell lines, individual correlation equations for each UFG were used to predict IC50s for each cell line based only on expression of that gene in that cell line. This was repeated for all UFGs and cell lines, followed by averaging predicted IC50s for each cell line based on either full sets or subsets of eribulin and vinorelbine UFGs. Predicted mean IC50s were then correlated with actual measured IC50s by linear regression analysis, thereby obtaining both statistical significance and the equation of the model.

Since IC50s predicted from equations of individual UFGs are actually drug-agnostic (independent of the cognate drug for which UFG status was accorded), MVR models built from eribulin and vinorelbine UFGs were assessed for their abilities to predict responses not just to the corresponding cognate drug but also to the 2 noncognate drugs (including paclitaxel). As expected, MVR models built with full sets of eribulin and vinorelbine UFGs showed highly significant correlations with their cognate drugs (Appendix A). Unexpectedly, however, full set MVR models also showed significant correlations with one or both noncognate drugs, presumably due to consolidation and statistical strengthening of nonsignificant numerical trends for noncognate drugs that can be observed for many individual UFGs (visible in Appendix A). Accordingly, MVR models using UFG subsets were constructed by prioritizing UFGs based on the highest individual correlation coefficients (R^2^ values) across all 100 cells lines for each gene. This approach yielded MVR models based on 4 eribulin UFGs (*C5ORF38*, *DAAM1*, *IRX2*, *CD70*) and 5 vinorelbine UFGs (*EPHA2*, *NGEF*, *SEPTIN10*, *TRIP10*, *VSIG10*), both of which showed highly significant correlations between predicted and measured IC50s across all 100 cell lines for cognate but not noncognate drugs.

## 3. Results

### 3.1. Antiproliferative Effects of Eribulin, Paclitaxel and Vinorelbine against 100 CCLE Cell Lines

In vitro antiproliferative potencies of eribulin, paclitaxel and vinorelbine against 100 CCLE cell lines were determined during 72 h compound exposures (Figure 1A–D, Table 1, Appendix A). Overall, eribulin IC50s correlated with those for paclitaxel and vinorelbine with high statistical significance across the 100 cells lines (Figure 1A), although slopes for paclitaxel and vinorelbine were notably shallower compared to eribulin, whose lowest IC50s dipped well below 10^−9^ M, thus driving the steeper eribulin slope. The overall order of potency was eribulin > paclitaxel > vinorelbine, with eribulin showing 1.5- to 6.7-fold greater potency compared to paclitaxel (means and medians, respectively), and 2.0- to 14.9-fold greater potency compared to vinorelbine (Table 1).

Although IC50s across the 100 cell lines were highly correlated as a whole, some lines individually responded differently to each drug, as shown by analysis of the most and least sensitive quartiles (25 cells lines with lowest and highest IC50s for each drug, respectively; Figure 1B–D). Thus, 5, 9 and 4 cell lines were uniquely associated with the most sensitive quartiles for eribulin, paclitaxel and vinorelbine, respectively, while 11 cell lines were shared by all 3 drugs, and 12 others were variously shared between the 3 remaining 2-drug pairs (Figure 1E). In the least sensitive quartiles, 7, 9 and 7 cell lines were unique to eribulin, paclitaxel and vinorelbine, respectively, with 12 cell lines shared by all 3 drugs and another 8 variously shared by the 3 remaining 2-drug pairs (Figure 1F).

### 3.2. Gene Expression Analysis of Most Sensitive versus Least Sensitive Cell Line Quartiles

Baseline expression data for 11,985 genes in each of the 100 CCLE cell lines were obtained from the publicly available DepMap database. Initial filtering was based on ratios of average expression of each gene across the 25 cell lines in the top (most sensitive) versus bottom (least sensitive) quartiles, defined separately for eribulin, paclitaxel and vinorelbine; top and bottom quartiles thus served as conceptual surrogates for responders and nonresponders, respectively. Z-score heat map analysis of gene expression at *p* < 0.05 stringency revealed visual groupings of genes that were positively or negatively associated with response to all three drugs, to each of the three possible drug pairs, or to each drug uniquely (Figure 2).

Volcano plots of gene expression fold-changes (fold-change refers to mean gene expression ratios between top and bottom quartiles, with ‘upregulated genes’ expressed at higher mean levels in the most sensitive versus least sensitive cell line quartile) versus *p*-value revealed 949, 646 and 1143 differentially expressed genes for eribulin, paclitaxel and vinorelbine at *p* < 0.05, and 61, 30 and 90 genes at *p* < 0.0025, respectively (Figure 3A–C and Appendix A). Further analysis showed that 342, 228 and 334 of the upregulated genes and 279, 189 and 392 of the downregulated genes were uniquely associated with responses to eribulin, paclitaxel and vinorelbine at *p* < 0.05, respectively (Figure 3D,E). With greater stringency at *p* < 0.0025, 34, 21 and 48 nonoverlapping upregulated genes and 18, 8 and 32 nonoverlapping downregulated genes were uniquely associated with eribulin, paclitaxel and vinorelbine responses, respectively (Figure 3F,G).

### 3.3. Identification of Unique Fingerprint Genes (UFGs) for Eribulin and Vinorelbine

While initial filtering by expression ratios between most and least sensitive cell line quartiles narrowed potentially relevant genes from many thousands to just a few dozen, this came at the expense of (i) not utilizing gene expression information from the 50 cell lines in the middle two quartiles, (ii) accepting a risk that small numbers of cell lines in the top and bottom quartiles might have outlier expression patterns that could disproportionately influence gene selection, and (iii) failing to provide a basis for identifying drug-specific UFGs based on the informational power available from all 100 cell lines. To address these limitations, a final curation of the 52, 29 and 80 *p* < 0.0025 genes for eribulin, paclitaxel and vinorelbine, respectively (Figure 3F,G), was performed to include only drug-unique genes that were at least 2-fold up- or downregulated between the most and least sensitive cell line quartiles (equivalent to absolute value of log_2_[fold-change] ≥ 1). Following this restriction, data from the middle two cell line quartiles were then reintroduced, and each gene in the three subsets was subjected to full linear regression analysis including expression data and IC50s from all 100 cell lines. Only genes with expression showing statistical significance for a given cognate drug but lacking statistical significance for the other two noncognate drugs were designated as UFGs for each drug. This selection process yielded 9 UFGs for eribulin and 13 UFGs for vinorelbine (Table 2; Appendix A). Unexpectedly, no gene, up- or downregulated, met these final criteria for designation as a paclitaxel UFG.

### 3.4. Molecular Interactions Associated with Eribulin and Vinorelbine UFG Sets

We next asked what mechanisms and pathways were associated with eribulin and vinorelbine UFGs. To this end, network propagation [19] analyses were performed to identify molecular networks associated with eribulin and vinorelbine UFGs followed by pathway enrichment analyses to identify the pathways highlighted by the resulting networks. Using the eribulin and vinorelbine UFG sets as query inputs, 100 gene networks (UFG interactomes) were obtained for both drugs (Figure 4A,B and Appendix A)—network propagation and Reactome analyses were not performed for paclitaxel since no gene met all criteria for a paclitaxel UFG. These networks included seven of nine eribulin UFGs and 13 of 13 vinorelbine UFGs; two UFGs for eribulin, *C5ORF38* and *GPR157*, were not themselves captured by network propagation. Interestingly, despite sharing mechanistic similarity as MTAs, only four genes overlapped between the eribulin and vinorelbine 100-gene propagated networks (*ESR2*, *HNRNPL*, *MTNR1B*, *TRIM25*). We speculate that this may relate to the fact that the original UFGs were selected, in part, based on lack of correlations to noncognate drugs.

### 3.5. Reactome Pathways Associated with Eribulin and Vinorelbine Response

To further elucidate the pathways associated with the eribulin and vinorelbine UFG interactomes, the eribulin and vinorelbine 100-gene propagated networks were used as query inputs for Reactome pathway enrichment analyses. The resulting Reactome maps indicate that most pathways for both drugs fall into three main Reactome groupings or ‘islands’: the Signal Transduction, Immune System and Cell Cycle islands (Figure 4C,D). While top-level viewing reveals these commonalities, detailed blowups of the three islands show clear differences in the specific pathways identified for the two drugs (Figure 4E–J). For instance, in the Signal Transduction island, the eribulin UFG interactome is strongly associated with Rho GTPase signaling pathways, while the vinorelbine interactome shows stronger associations with FGFR, EGFR/ERBB and NGF pathways (Figure 4E,F). Similarly, in the Immune System island, the eribulin UFG interactome shows strong association with Toll-like receptor (TLR) signaling, which notably was not covered at all by vinorelbine, while the vinorelbine UFG interactome shows stronger associations with both adaptive and innate immune branches as well as cytokine signaling (Figure 4G,H). Finally, while Cell Cycle island pathways were generally more similar for the eribulin and vinorelbine UFG interactomes compared to Signal Transduction and Immune System islands, eribulin alone was associated with chromosome maintenance pathways, while vinorelbine alone was associated with cell cycle checkpoint pathways (Figure 4I,J). Thus, despite top-level Reactome island commonalities between the two drugs, detailed inspection within the islands themselves reveals significant differences in specific pathway associations.

Cataloging of the individual Reactome pathways identified confirms differences in pathway involvement between the UFGs of the two drugs. Using combined criteria of *p* < 0.05 significance plus q < 0.1 to account for FDR, 26 and 122 Reactome pathways were identified for eribulin and vinorelbine UFG interactomes, respectively (Table 3); considerable differences in pathways are evident among these. First, only two Reactome pathways were shared: Immune System and Signaling by Interleukins. More notably, TLR-related pathways accounted for 17/26 (65.4%) pathways for eribulin, yet none of the 122 pathways for vinorelbine. Seven of the remaining nine pathways for eribulin involved Rho GTPase- and cell cycle/mitosis-related pathways, which were all but absent for vinorelbine. In contrast, 45 of 122 (36.9%) Reactome pathways for vinorelbine involved FGFR, PI3K/Akt and EGFR/ERBB, yet such pathways were not highlighted by the eribulin UFG interactome. Thus, cataloging Reactome pathways in Table 3 confirms the visual observations of Figure 4 that, despite top-level commonalities seen for the Signal Transduction, Immune System and Cell Cycle islands, individual pathway details point to considerable differences in pathway determinants of eribulin and vinorelbine response, as highlighted by the UFG interactomes.

### 3.6. Multivariate Regression (MVR) Model Building to Predict Drug Sensitivities

Finally, multigene MVR models for eribulin and vinorelbine were assembled (not done for paclitaxel since no gene qualified as a paclitaxel UFG) by first calculating each cell line’s predicted IC50 based on individual expression of each of the 9 eribulin or 13 vinorelbine UFGs for that cell line, averaging the resulting 9 or 13 predicted IC50s for each cell line, then comparing the averaged predicted IC50s to actual measured IC50s for each cell line for all three drugs. Although no gene met the criteria for assignment as a paclitaxel UFG, predicted IC50s based on expression of eribulin or vinorelbine UFGs are conceptually drug agnostic, so paclitaxel was included as a drug specificity comparator. Not surprisingly, the resulting MVR models built from all 9 or 13 eribulin or vinorelbine UFGs, respectively, successfully predicted cognate drug IC50s with high statistical significance (Appendix A). Unexpectedly, however, these two MVR models also showed statistical correlations with one or both noncognate drugs (albeit at lower statistical significance), even though one criterion for each individual UFG was lack of statistical correlation with the two noncognate drugs. Thus, IC50s predicted from the 9 UFG eribulin MVR model correlated with measured IC50s not just for eribulin, but also for paclitaxel and vinorelbine, with *p*-values of <0.001, 0.022 and 0.021, respectively, while those predicted from the 13 UFG vinorelbine MVR model correlated with measured IC50s for not just vinorelbine, but also for eribulin with *p*-values of <0.001 and 0.009, respectively; correlation with paclitaxel was not significant (*p* = 0.097; Appendix A). We speculate that acquisition of statistical significance for noncognate drugs in MVR models built from all 9 eribulin or 13 vinorelbine UFGs probably reflects consolidation and statistical strengthening of nonstatistical numeric trends for noncognate drugs that can be seen for several individual UFGs (Appendix A).

The use of smaller UFG subsets was then employed to generate MVR models that could predict IC50s for cognate but not noncognate drugs. Prioritizing UFGs by highest correlation coefficients (individual R^2^ values) yielded a four-gene eribulin MVR model (*C5ORF38*, *DAAM1*, *IRX2*, *CD70*) and a five-gene vinorelbine MVR model (*EPHA2*, *NGEF*, *SEPTIN10*, *TRIP10*, *VSIG10*); predicted IC50s from both models correlated with measured IC50s across all 100 cell lines with high statistical significance for cognate but not noncognate drugs (Figure 5A,C). Equations for these MVR models are as follows, with gene expression values in RPKM and predicted IC50 values in log_10_[M]:

Four-gene eribulin MVR model:IC50_predicted_ = (*CD70*/5.40) − (*C5ORF38*/3.21) − (*DAAM1*/2.06) − (*IRX2*/3.30) − 8.54

Five-gene vinorelbine MVR model:IC50_predicted_ = (*EPHA2*/11.55) + (*NGEF*/7.75) + (*SEPTIN10*/8.40) + (*TRIP10*/4.25) + (*VSIG10*/4.49) − 9.81

To determine if these MVR models could form the basis for predictive biomarker gene panels, we returned to the starting concept of most versus least sensitive cell line quartiles as surrogates for “likely responders” versus “likely nonresponders,” respectively. As shown in Figure 5B,D, both MVR models calculated mean IC50s for the most and least sensitive cell line quartiles with high accuracy relative to actual measured IC50s (Figure 5B,D).

In summary, an approach defining drug-specific top and bottom cell line quartiles from 100 CCLE cell lines allowed distillation of 11,985 genes down to 52 and 80 drug-nonoverlapping *p* < 0.0025 genes for eribulin and vinorelbine, respectively, which were then further distilled to 9 and 13 UFGs by setting expression ratio and drug specificity thresholds followed by reintroducing data from all 100 cell lines. MVR models comprised of four- and five-gene UFG subsets accurately calculated mean eribulin and vinorelbine IC50s of the original top and bottom cell line quartiles based solely on baseline gene expression levels. We propose that the four- and five-gene MVR panels identified here may form the basis for drug-specific predictive biomarkers for eribulin and vinorelbine response, respectively.

## 4. Discussion

Despite decades of use in cancer therapy, reliable biomarkers to predict response to clinically approved MTAs remain lacking. While all MTAs share the top-level mechanism of inhibiting MT dynamics, differences in binding sites on α/β-tubulin monomers, locations of those binding sites in the context of polymerized MTs, net effects as MT stabilizers versus MT destabilizers and demonstrable differences in clinical profiles indicate that despite our extensive knowledge of MTA biochemistry, utilizing such knowledge to create predictive biomarkers remains elusive. To address this, we coupled measured response data from 100 human cancer cell lines from the CCLE with gene expression data for almost 12,000 genes from each cell line to identify molecular and pathway correlates of eribulin response, contrasting results with two other clinically used MTAs, paclitaxel and vinorelbine. Our results identified nine genes (termed UFGs) whose expression is uniquely associated with response to eribulin versus the other two drugs, with four eribulin UFGs being sufficient for a multigene panel that accurately predicts high versus low eribulin response. Separately, 13 UFGs were identified as uniquely associated with vinorelbine response, with 5 of these being sufficient for a predictive vinorelbine multigene panel. Notably, no genes uniquely associated with paclitaxel response were identified using the same stringent criteria used to identify eribulin and vinorelbine UFGs.

In addition to its cytotoxic antimitotic activity [5,6,7], eribulin is unusual among MTAs in that it also exerts a wide range of noncytotoxic effects on the tumor microenvironment, including vascular remodeling or normalization resulting in increased tumor perfusion and mitigation of hypoxia, phenotypic changes including reversal of EMT and induced cellular differentiation, and effects on the tumor immune microenvironment [8,9,10,11,12,22,23]. We hypothesized that the existence of such nonmitotic effects might invoke additional cellular pathways that could provide the basis for a gene expression-based biomarker strategy specific for eribulin.

Responses of 100 CCLE human cancer cell lines to eribulin and comparator MTAs paclitaxel and vinorelbine were first quantified by IC50 values and ordered by response. While sensitivities across the 100 cell lines trended in the same direction for the three drugs, the range of eribulin’s response from most to least sensitive was considerably greater than the other two drugs, reflected by a much steeper IC50 slope for eribulin compared to the shallower and parallel slopes for paclitaxel and vinorelbine. Importantly, while IC50s for the least sensitive cell lines were in the same 10^−7^ to 10^−6^ M range for all three drugs, eribulin’s steeper slope resulted primarily from increased responses in its most sensitive cell lines, which collectively dropped into the sub-nM IC50 range. In addition, several nonoverlapping cell lines existed for each drug in their respective top and bottom quartiles, suggesting that aggregating response data across large numbers of cell lines might yield information on drug-specific pathways and mechanisms, a concept first established by the well-known NCI60 cell line screen (https://dtp.cancer.gov/discovery_development/nci-60/; accessed on 26 July 2022). Coincidentally, the NCI60 screen was first used to predict a tubulin-based mechanism for eribulin’s natural product parent, halichondrin B [24,25]. We speculated that the enhanced responsiveness of many cell lines to eribulin relative to paclitaxel and vinorelbine resulted from specific upregulation of pathways’ governing response, as opposed to general downregulation of resistance pathways that might explain the response to paclitaxel and vinorelbine. Such observations supported the concept that eribulin’s unique biology among MTAs might provide a basis for developing a predictive eribulin biomarker strategy.

Using top and bottom quartiles as surrogates for responders and nonresponders, respectively, IC50s were correlated with DepMap gene expression data for 11,985 genes. Z-score heat mapping and filtering on *p*-value and gene expression ratios between top and bottom quartiles provided the first indications that the three drugs could be distinguished from each other for biomarker development. Notably, at *p* < 0.0025 stringency, 52 and 80 genes were unique to eribulin and vinorelbine, respectively, with no genes shared between eribulin and paclitaxel, and only nine genes shared between eribulin and vinorelbine. At the same stringency, vinorelbine and paclitaxel shared only one gene. Thus, paclitaxel, a microtubule stabilizer, showed the least similarity with eribulin and vinorelbine, both microtubule destabilizers, yet even eribulin and vinorelbine could be distinguished from each other based on many more genes uniquely associated with each drug compared to shared genes between them.

Further restriction of the *p* < 0.0025 genes to include only those with at least 2-fold expression difference between top and bottom quartiles, followed by reintroduction of data from the 50 cell lines in the middle two quartiles, resulted in identification of 9 and 13 UFGs for eribulin and vinorelbine, respectively. Interestingly, no gene met these final UFG criteria for paclitaxel. Notably, seven of the nine eribulin UFGs were upregulated with increased response, while 11 of the 13 vinorelbine UFGs were downregulated with increased response, supporting our hypothesis that increased eribulin response may be driven by upregulation of pathways governing sensitivity, whereas increased vinorelbine response may depend more on downregulation of general resistance pathways.

To elucidate biological pathways associated with eribulin and vinorelbine UFGs, network propagation [19] was used to identify networks highlighted by the 9 and 13 UFG sets into expanded 100-gene networks, which were then used as query inputs for Reactome pathway enrichment analyses. Intriguingly, although eribulin and vinorelbine are both microtubule-depolymerizing MTAs, only 4 of 100 genes in the propagated networks for each drug overlapped, likely because assignment as a UFG excluded genes that correlated with response to noncognate drugs. Reactome pathway analyses using the 100-gene propagated networks as query inputs revealed similar top-level pathway commonalities for both drugs within the major Signal Transduction, Immune System and Cell Cycle Reactome islands. Nevertheless, specific pathways identified within these islands differed considerably for the two drugs. Thus, eribulin showed strongest associations with TLR- and Rho GTPase-associated signaling pathways, while vinorelbine pathways emphasized FGFR, EGFR/ERBB and both adaptive and innate immune signaling. Cataloging of the 26 and 122 individual Reactome pathways identified for eribulin and vinorelbine UFG interactomes, respectively, confirmed the lack of specific pathway overlaps. For instance, TLR-related pathways accounted for ~65% (17/26) of Reactome pathways for eribulin, yet none for vinorelbine. On the other hand, ~37% (45/122) of Reactome pathways for vinorelbine involved FGFR, PI3K/Akt and EGFR/ERBB, yet such pathways were completely absent for eribulin. While further work will be required to determine how the identified pathways govern responses to eribulin and vinorelbine, results at the levels of individual UFGs, 100-gene propagated networks and individual Reactome pathways strongly suggest that despite their shared classification as MT depolymerizing MTAs, responses to eribulin and vinorelbine are governed by different genetic and pathway determinants and can be readily distinguished from each other by gene expression patterns.

Finally, subsets of eribulin and vinorelbine UFGs prioritized by highest individual correlation coefficients were used to build drug-specific MVR models that correlate with responses across all 100 cell lines and accurately predict average response levels of the most and least sensitive quartiles based solely on expression levels of only four or five genes, respectively. While others have reported gene or pathway signatures for MTAs, these have been mainly for paclitaxel [26,27,28,29,30,31] and have not, to our knowledge, explored the rigorous drug specificity within MTAs required by our criteria. In this regard, it is noteworthy that no gene met all UFG criteria for paclitaxel, especially considering the importance of cellular signaling to both eribulin and vinorelbine responsiveness. While speculative, paclitaxel may represent a ‘purer’ cytotoxic based on its internal MT lumen binding site [32,33], which may prevent interactions with microtubule-associated proteins (MAPs) involved in cellular signaling. In contrast, vinorelbine and eribulin bind at or near, respectively, the so-called β-tubulin vinca binding site near MT ends [33,34,35], where such binding may more readily interfere with MAP-mediated signaling events. In this context, eribulin binds almost exclusively to exposed β-tubulin at growing MT plus (+) ends [34,35], and such binding rapidly blocks association of plus-end binding proteins (+TIPS) such as EB1 [36,37] that serve as scaffolds for multiprotein assemblies involved in both structural and signaling activities [38,39,40]. For instance, eribulin binding to MT (+) ends disrupts p130Cas/Src interactions, leading to cortical localization of E-cadherin [41], a hallmark of eribulin’s ability to reverse EMT [8]. In addition, both eribulin and vinorelbine binding rapidly inhibits TGFβ-induced Smad signaling by preventing MT-dependent Smad2/3 transport into the nucleus [42]. Thus, binding of both eribulin and vinorelbine to externally accessible sites at or near MT ends compared to paclitaxel’s inner MT luminal binding may help explain both the failure to identify drug-specific paclitaxel UFGs as well as the observation that cellular signaling themes dominated for both eribulin and vinorelbine.

Considering the importance of signaling-related Reactome pathways in eribulin and vinorelbine responses, it is reasonable to ask if the four eribulin and five vinorelbine UFGs in the predictive MVR models directly control responses to the two drugs. Strictly speaking, UFG expression levels were only correlated with response, so firm conclusions of direct involvement cannot be drawn. However, with that caveat, information on known roles of the genes permits some speculation. For eribulin, *IRX2* is a homeobox gene important in normal embryonic development and has been implicated in cancer, while *C5ORF38* is coordinately regulated with *IRX2*, suggesting both are involved in cellular differentiation and growth regulation [43]. Similarly, *DAAM1* is involved in Wnt signaling and early embryonic gastrulation, both processes associated with cellular differentiation [44]. Finally, *CD70* codes for a ligand for the immune costimulatory molecule CD27 and thus is involved in the activation and proliferation of T cells, including regulatory T cells [45]. Thus, while speculative, all four of the eribulin UFGs in the predictive MVR model appear related to cell differentiation-, activation- or proliferation-related processes.

For vinorelbine, *EPHA2* codes for EPH receptor A2, a tyrosine kinase involved in cancer-related signaling, while *NGEF* codes for a guanine nucleotide exchange factor associated with signaling from EPH receptor A2, RhoA, Rac1 and CDC42 as well as cellular transformation and tumorigenesis [46,47]. *SEPTIN10* is associated with B cell leukemias [48], while *TRIP10* is involved in tumorigenesis and cancer progression [49]. Finally, *VSIG10* codes for an immunoglobulin-related protein associated with both cell adhesion and macrophage involvement in colonic pathologies including colon cancer [50]. Thus, while solid mechanistic links between drug response and the UFGs in the eribulin and vinorelbine MVR models cannot be established by purely correlational data, the known roles of these genes are consistent with potential mechanisms that set response sensitivity levels to these two drugs.

## 5. Conclusions

In summary, we have combined response data for eribulin, paclitaxel and vinorelbine from 100 human cancer cell lines together with gene expression data to identify small numbers of genes (UFGs) that are uniquely associated with eribulin and vinorelbine responses. Reactome pathway analyses based on UFG-seeded propagated gene networks revealed that responses to both eribulin and vinorelbine are dominantly associated with cellular signaling processes as opposed to canonical MT-based antimitotic processes, perhaps also explaining our inability to identify paclitaxel UFGs using the same criteria. Despite top-level dependence on cellular signaling for response to both eribulin and vinorelbine, detailed analyses of the specific Reactome pathways identified reveal considerable discrimination between the two drugs. Finally, we show that small subsets of eribulin and vinorelbine UFGs can be successfully combined into MVR models that accurately predict a high versus low response to the two drugs based on expression of only four or five genes. Our results indicate that further investigation of the genes, pathways and MVR panels identified here is warranted, with further validation in both preclinical tumor models and in the clinical setting being of highest priority. Ultimately, our hope is that the current results together with such future validation work will lead to the development of drug-specific, gene expression-based predictive biomarker panels for eribulin and vinorelbine that support improved therapeutic decision-making in clinical settings.

## Figures and Tables

**Figure 1 cancers-14-04532-f001:**
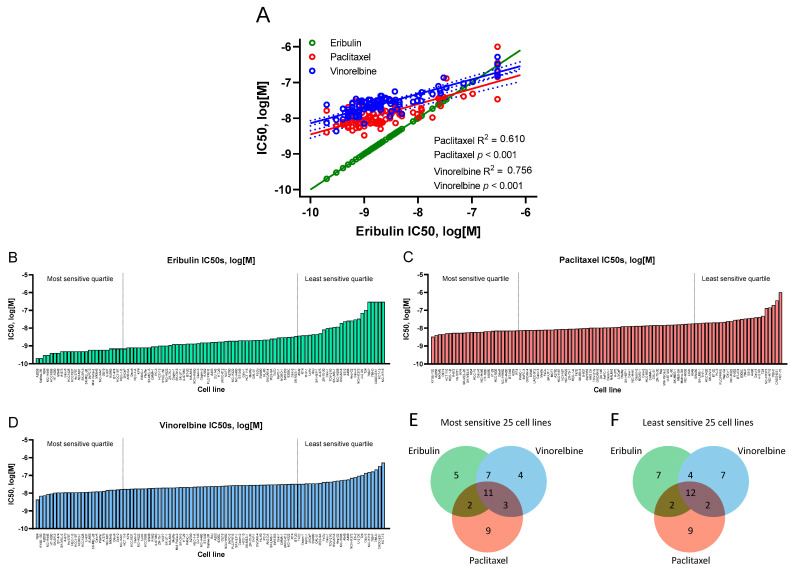
Antiproliferative effects of eribulin, paclitaxel and vinorelbine on 100 CCLE cell lines. (**A**) Correlations of IC50s for eribulin versus paclitaxel and vinorelbine, with eribulin versus itself shown to compare slopes with paclitaxel and vinorelbine (R^2^ and *p*-values not shown for eribulin versus itself). For eribulin (**B**), paclitaxel (**C**) and vinorelbine (**D**), cell lines are ordered from most sensitive (left) to least sensitive (right), with IC50s as log[M]. The most and least sensitive quartiles for each drug are delineated by vertical dashed lines. Measured IC50s for 5 eribulin and 1 paclitaxel cell lines exceeded the highest concentrations tested for these 2 drugs; see Material and Methods for correction strategies used. Panels (**E**,**F**) show Venn diagrams of numbers of unique and overlapping cell lines in the most and least sensitive cell line quartiles for each drug.

**Figure 2 cancers-14-04532-f002:**
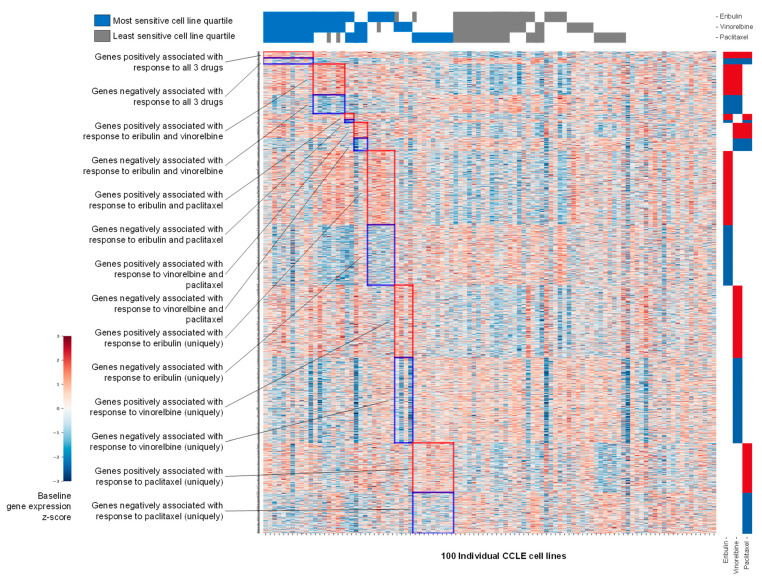
Z-score heat map showing baseline gene expression across 100 CCLE cell lines. Genes shown (rows) are those positively or negatively associated at *p* < 0.05 with the most sensitive versus least sensitive cell line quartiles for eribulin, vinorelbine and paclitaxel as indicated across the top of the heat map. The left and right sides of the *y*-axis show textual or visual representations of associations of genes with 1, 2 or all 3 of the tested drugs.

**Figure 3 cancers-14-04532-f003:**
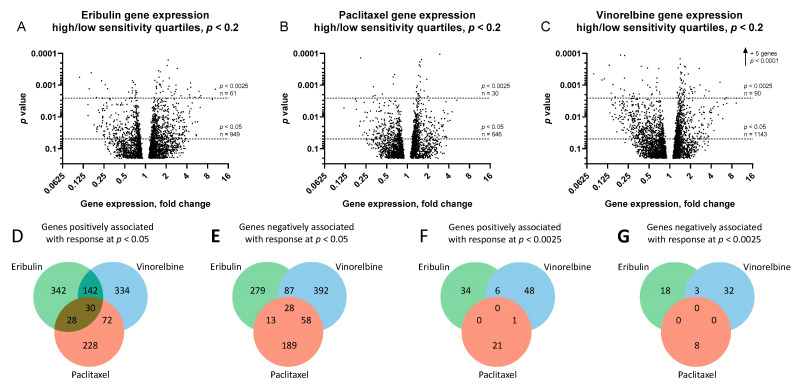
Genes associated with responses to eribulin, paclitaxel and vinorelbine. Volcano plots of ratios of gene expression levels between most and least sensitive cell line quartiles are plotted versus *p*-values for eribulin (**A**), paclitaxel (**B**) and vinorelbine (**C**). For visual clarity, only genes with *p* < 0.2 are plotted (3149, 2486 and 3435 genes for eribulin, paclitaxel and vinorelbine, respectively). Horizontal dashed lines denote statistical significance and numbers of genes below *p* < 0.05 and *p* < 0.0025. Venn diagrams of numbers of genes positively (**D**,**F**) and negatively (**E**,**G**) associated with responses to eribulin, paclitaxel and vinorelbine at *p* < 0.05 (**D**,**E**) and *p* < 0.0025 (**F**,**G**) stringency levels are shown.

**Figure 4 cancers-14-04532-f004:**
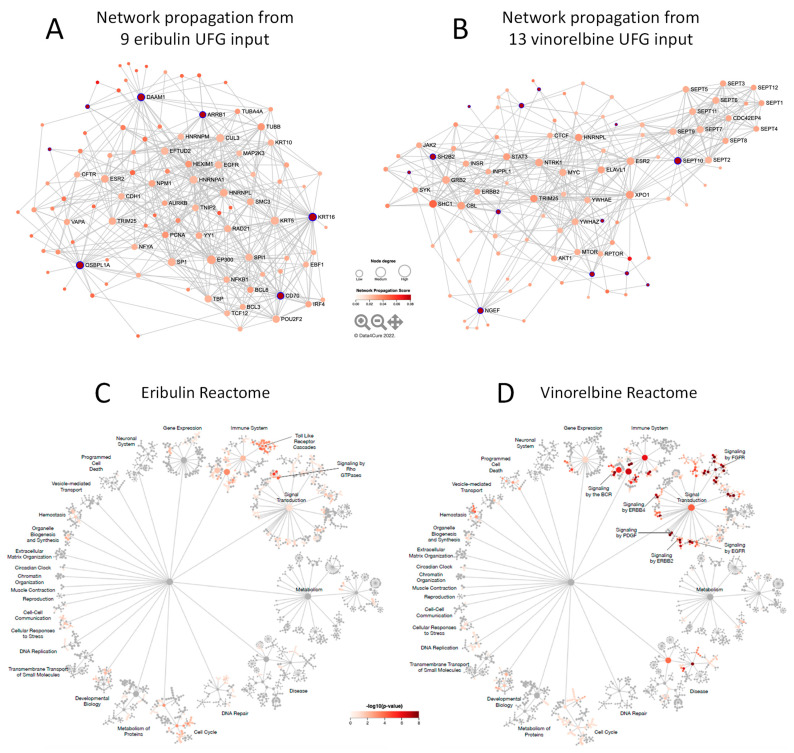
Network propagation and Reactome pathway analyses. (**A**,**B**) Network propagation from query sets consisting of the 9 eribulin UFGs (**A**) and the 13 vinorelbine UFGs (**B**) yielded the 100-gene networks shown, which included 7 of the 9 eribulin UFGs and all 13 of the vinorelbine UFGs; 2 eribulin UFGs, *C5ORF38* and *GPR157*, were not themselves captured in the network propagation. Purple-highlighted network nodes represent original UFG query input genes. (**C**–**J**) Reactome pathways derived from the 100-gene query sets obtained from network propagation. Full Reactome maps are shown for eribulin (**C**) and vinorelbine (**D**), with blow ups of the Signal Transduction (**E**,**F**), Immune System (**G**,**H**) and Cell Cycle (**I**,**J**) islands shown for eribulin (**E**,**G**,**I**) and vinorelbine (**F**,**H**,**J**).

**Figure 5 cancers-14-04532-f005:**
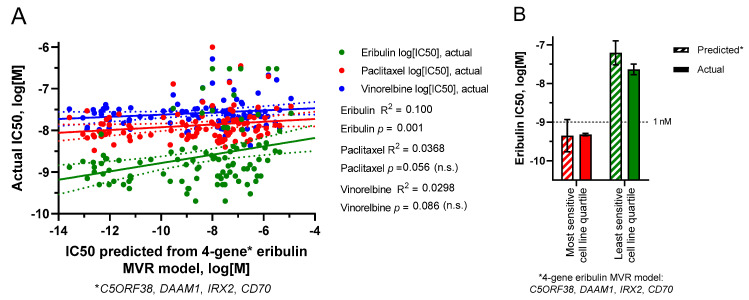
Predicted versus actual IC50s based on limited gene MVR models. (**A**,**B**) predicted IC50s were derived from a 4-gene subset (*C5ORF38*, *DAAM1*, *IRX2*, *CD70*) of the 9 eribulin UFGs. (**C**,**D**) predicted IC50s were derived from a 5-gene subset (*EPHA2*, *NGEF*, *SEPTIN10*, *TRIP10*, *VSIG10*) of the 13 vinorelbine UFGs. (**A**,**C**) linear regressions of predicted versus actual IC50s for all 100 cell lines. (**B**,**D**) predicted versus actual mean IC50s (±SEM) for the most sensitive and least sensitive cell line quartiles as defined in Figure 1. For (**A****,C**), linear regression lines (solid) are shown together with their corresponding 95% confidence bands (dotted). For visual comparison, scales of x- and y-axes were kept the same between both panels here and Panels (**A****,B**) of Appendix A. No statistically significant differences were found between predicted and actual quartile mean IC50s in (**B**) or (**D**).

**Table 1 cancers-14-04532-t001:** Antiproliferative activities (IC50s) against 100 CCLE cell lines ^1^.

Compound	Mean, nM	SD	SEM	Median, nM
Eribulin ^2^	20.1	66.2	6.6	1.6
Paclitaxel ^2^	31.0	106.9	10.7	10.7
Vinorelbine	39.9	65.2	6.5	23.9

^1^ Averaged values for all 100 cell lines are presented here. Individual IC50s for each drug for all 100 cell lines are presented in Appendix A. ^2^ See Material and Methods for strategy used to address the 5 eribulin and 1 paclitaxel cell lines whose IC50s exceeded the highest concentrations tested for these 2 drugs.

**Table 2 cancers-14-04532-t002:** Unique fingerprint genes (UFGs) associated with drug response ^1,2,3^.

Eribulin	Vinorelbine
Up (7): *ARRB1*, *C5ORF38*, *DAAM1*, *GPR157*, *IRX2*, *KRT16*, *OSBPL1A*	Up (2): *PREX1*, *SH2B2*
Down (2): *BICC1*, *CD70*	Down (11): *EPHA2*, *GSTT2*, *GSTT2B*, *NGEF*, *PEAR1*, *PRSS3*, *RAP1GAP2*, *SEPTIN10*, *STEAP2*, *TRIP10*, *VSIG10*

^1^ See Materials and Methods and Results text for criteria and methods used to define and identify UFGs. Individual linear regression plots of gene expression versus IC50s across all 100 cell lines for each of the 22 genes in this table, including R^2^ and *p*-values, are presented in Appendix A. ^2^ No gene met all criteria for paclitaxel, so only eribulin and vinorelbine UFGs are listed here. ^3^ Genes are listed in alphabetical order within each subgroup. Up and down genes refer to higher and lower gene expression levels, respectively, associated with greater drug response (lower IC50s).

**Table 3 cancers-14-04532-t003:** Reactome pathways associated with eribulin and vinorelbine response at *p* < 0.05 plus q < 0.1 ^1^.

Drug	Reactome Pathway ^2^	*p*-Value	q-Value	Genes in Pathway	Over-Lapping Genes	Shared ^3^
Eribulin	RHO GTPase Effectors	0.000010	0.012839	251	10	
Eribulin	RHO GTPases Activate Formins	0.000016	0.012839	114	7	
Eribulin	Signaling by Rho GTPases	0.000045	0.019422	363	11	
Eribulin	Toll Like Receptor 9 (TLR9) Cascade	0.000047	0.019422	92	6	
Eribulin	MyD88 cascade initiated on plasma membrane	0.000343	0.057053	85	5	
Eribulin	Toll Like Receptor 10 (TLR10) Cascade	0.000343	0.057053	85	5	
Eribulin	Toll Like Receptor 5 (TLR5) Cascade	0.000343	0.057053	85	5	
Eribulin	TRAF6 mediated induction of NFkB and MAP kinases upon TLR7/8 or 9 activation	0.000363	0.057053	86	5	
Eribulin	MyD88 dependent cascade initiated on endosome	0.000403	0.057053	88	5	
Eribulin	Toll Like Receptor 7/8 (TLR7/8) Cascade	0.000403	0.057053	88	5	
Eribulin	Toll-Like Receptors Cascades	0.000514	0.057053	142	6	
Eribulin	MyD88: Mal cascade initiated on plasma membrane	0.000574	0.057053	95	5	
Eribulin	Toll Like Receptor 2 (TLR2) Cascade	0.000574	0.057053	95	5	
Eribulin	Toll Like Receptor TLR1:TLR2 Cascade	0.000574	0.057053	95	5	
Eribulin	Toll Like Receptor TLR6:TLR2 Cascade	0.000574	0.057053	95	5	
Eribulin	Immune System	0.000655	0.057053	1232	19	X
Eribulin	MAP kinase activation in TLR cascade	0.000674	0.057053	56	4	
Eribulin	MyD88-independent TLR3/TLR4 cascade	0.000694	0.057053	99	5	
Eribulin	Toll Like Receptor 3 (TLR3) Cascade	0.000694	0.057053	99	5	
Eribulin	TRIF-mediated TLR3/TLR4 signaling	0.000694	0.057053	99	5	
Eribulin	Cell Cycle	0.001075	0.083265	523	11	
Eribulin	Cell Cycle, Mitotic	0.001134	0.083265	445	10	
Eribulin	Signaling by Interleukins	0.001164	0.083265	111	5	X
Eribulin	Activated TLR4 signalling	0.001312	0.086320	114	5	
Eribulin	G2/M Transition	0.001312	0.086320	114	5	
Eribulin	Mitotic G2-G2/M phases	0.001418	0.089698	116	5	
Vinorelbine	Signalling by NGF	4.383506 × 10^−12^	7.210868 × 10^−9^	288	18	
Vinorelbine	Signaling by SCF-KIT	6.752605 × 10^−11^	5.554018 × 10^−8^	144	13	
Vinorelbine	Signaling by FGFR3	3.228850 × 10^−10^	8.992552 × 10^−8^	163	13	
Vinorelbine	Signaling by FGFR4	3.228850 × 10^−10^	8.992552 × 10^−8^	163	13	
Vinorelbine	Signaling by FGFR1	3.485981 × 10^−10^	8.992552 × 10^−8^	164	13	
Vinorelbine	DAP12 signaling	3.761623 × 10^−10^	8.992552 × 10^−8^	165	13	
Vinorelbine	Signaling by FGFR2	4.056969 × 10^−10^	8.992552 × 10^−8^	166	13	
Vinorelbine	Signaling by FGFR	4.373277 × 10^−10^	8.992552 × 10^−8^	167	13	
Vinorelbine	NGF signalling via TRKA from the plasma membrane	5.528565 × 10^−10^	1.010499 × 10^−7^	207	14	
Vinorelbine	DAP12 interactions	1.111051 × 10^−9^	1.661526 × 10^−7^	180	13	
Vinorelbine	Signaling by EGFR	1.111051 × 10^−9^	1.661526 × 10^−7^	180	13	
Vinorelbine	Signaling by the B Cell Receptor (BCR)	1.309948 × 10^−9^	1.673749 × 10^−7^	221	14	
Vinorelbine	Downstream signaling of activated FGFR1	1.627962 × 10^−9^	1.673749 × 10^−7^	150	12	
Vinorelbine	Downstream signaling of activated FGFR2	1.627962 × 10^−9^	1.673749 × 10^−7^	150	12	
Vinorelbine	Downstream signaling of activated FGFR3	1.627962 × 10^−9^	1.673749 × 10^−7^	150	12	
Vinorelbine	Downstream signaling of activated FGFR4	1.627962 × 10^−9^	1.673749 × 10^−7^	150	12	
Vinorelbine	Interleukin-3, 5 and GM-CSF signaling	1.767887 × 10^−9^	1.710691 × 10^−7^	45	8	
Vinorelbine	Signaling by ERBB4	2.205662 × 10^−9^	2.015730 × 10^−7^	154	12	
Vinorelbine	Downstream signal transduction	3.945920 × 10^−9^	3.416336 × 10^−7^	162	12	
Vinorelbine	Signaling by ERBB2	4.540573 × 10^−9^	3.734621 × 10^−7^	164	12	
Vinorelbine	PI3K/AKT activation	6.543594 × 10^−9^	5.125815 × 10^−7^	103	10	
Vinorelbine	Diseases of signal transduction	1.515872 × 10^−8^	1.133459 × 10^−6^	267	14	
Vinorelbine	Signaling by PDGF	1.780926 × 10^−8^	1.273749 × 10^−6^	185	12	
Vinorelbine	Role of LAT2/NTAL/LAB on calcium mobilization	2.254580 × 10^−8^	1.545327 × 10^−6^	151	11	
Vinorelbine	Downstream signaling events of B Cell Receptor (BCR)	7.714714 × 10^−8^	4.001226 × 10^−6^	170	11	
Vinorelbine	PI-3K cascade:FGFR1	7.783541 × 10^−8^	4.001226 × 10^−6^	100	9	
Vinorelbine	PI-3K cascade:FGFR2	7.783541 × 10^−8^	4.001226 × 10^−6^	100	9	
Vinorelbine	PI-3K cascade:FGFR3	7.783541 × 10^−8^	4.001226 × 10^−6^	100	9	
Vinorelbine	PI-3K cascade:FGFR4	7.783541 × 10^−8^	4.001226 × 10^−6^	100	9	
Vinorelbine	PI3K events in ERBB2 signaling	7.783541 × 10^−8^	4.001226 × 10^−6^	100	9	
Vinorelbine	PI3K events in ERBB4 signaling	7.783541 × 10^−8^	4.001226 × 10^−6^	100	9	
Vinorelbine	PIP3 activates AKT signaling	7.783541 × 10^−8^	4.001226 × 10^−6^	100	9	
Vinorelbine	GAB1 signalosome	1.096689 × 10^−7^	5.466829 × 10^−6^	104	9	
Vinorelbine	Fc epsilon receptor (FCERI) signaling	1.417907 × 10^−7^	6.860169 × 10^−6^	223	12	
Vinorelbine	Immune System	1.922234 × 10^−7^	8.825209 × 10^−6^	1232	27	X
Vinorelbine	Signaling by Interleukins	1.931353 × 10^−7^	8.825209 × 10^−6^	111	9	X
Vinorelbine	PI3K/AKT Signaling in Cancer	3.077967 × 10^−7^	0.000014	85	8	
Vinorelbine	Innate Immune System	8.845427 × 10^−7^	0.000038	689	19	
Vinorelbine	Adaptive Immune System	2.458655 × 10^−6^	0.000104	665	18	
Vinorelbine	Cytokine Signaling in Immune system	4.429201 × 10^−6^	0.000182	308	12	
Vinorelbine	CD28 costimulation	4.691035 × 10^−6^	0.000187	32	5	
Vinorelbine	Regulation of mRNA stability by proteins that bind AU-rich elements	4.785766 × 10^−6^	0.000187	86	7	
Vinorelbine	Regulation of KIT signaling	6.232763 × 10^−6^	0.000238	16	4	
Vinorelbine	Insulin receptor signalling cascade	8.078116 × 10^−6^	0.000302	93	7	
Vinorelbine	Regulation of signaling by CBL	0.000010	0.000379	18	4	
Vinorelbine	CD28 dependent PI3K/Akt signaling	0.000020	0.000711	21	4	
Vinorelbine	HuR stabilizes mRNA	0.000026	0.000922	8	3	
Vinorelbine	Constitutive Signaling by AKT1 E17K in Cancer	0.000035	0.001189	24	4	
Vinorelbine	Signaling by Insulin receptor	0.000036	0.001223	117	7	
Vinorelbine	GPVI-mediated activation cascade	0.000040	0.001291	49	5	
Vinorelbine	Interleukin-2 signaling	0.000040	0.001291	49	5	
Vinorelbine	SHC-related events	0.000041	0.001299	25	4	
Vinorelbine	VEGFR2 mediated vascular permeability	0.000048	0.001497	26	4	
Vinorelbine	Signal Transduction	0.000055	0.001666	2260	33	
Vinorelbine	Integrin alphaIIb beta3 signaling	0.000056	0.001683	27	4	
Vinorelbine	Interleukin receptor SHC signaling	0.000065	0.001917	28	4	
Vinorelbine	IRS-related events	0.000070	0.002024	89	6	
Vinorelbine	Interleukin-6 signaling	0.000076	0.002165	11	3	
Vinorelbine	IGF1R signaling cascade	0.000090	0.002460	93	6	
Vinorelbine	Signaling by Type 1 Insulin-like Growth Factor 1 Receptor (IGF1R)	0.000090	0.002460	93	6	
Vinorelbine	Disease	0.000106	0.002860	714	16	
Vinorelbine	SHC1 events in ERBB2 signaling	0.000112	0.002969	32	4	
Vinorelbine	Signalling to RAS	0.000143	0.003724	34	4	
Vinorelbine	Signal attenuation	0.000166	0.004255	14	3	
Vinorelbine	Signalling to STAT3	0.000187	0.004720	3	2	
Vinorelbine	TP53 Regulates Metabolic Genes	0.000195	0.004720	68	5	
Vinorelbine	Transcriptional Regulation by TP53	0.000195	0.004720	68	5	
Vinorelbine	VEGFA-VEGFR2 Pathway	0.000195	0.004720	107	6	
Vinorelbine	Platelet Aggregation (Plug Formation)	0.000200	0.004756	37	4	
Vinorelbine	Constitutive Signaling by EGFRvIII	0.000206	0.004768	15	3	
Vinorelbine	Signaling by EGFRvIII in Cancer	0.000206	0.004768	15	3	
Vinorelbine	Costimulation by the CD28 family	0.000272	0.006221	73	5	
Vinorelbine	Signaling by VEGF	0.000289	0.006514	115	6	
Vinorelbine	Platelet activation, signaling and aggregation	0.000340	0.007564	221	8	
Vinorelbine	SHC activation	0.000372	0.008165	4	2	
Vinorelbine	Signalling to ERKs	0.000393	0.008513	44	4	
Vinorelbine	Constitutive Signaling by Ligand-Responsive EGFR Cancer Variants	0.000428	0.008920	19	3	
Vinorelbine	Signaling by EGFR in Cancer	0.000428	0.008920	19	3	
Vinorelbine	Signaling by Ligand-Responsive EGFR Variants in Cancer	0.000428	0.008920	19	3	
Vinorelbine	G beta:gamma signalling through PI3Kgamma	0.000550	0.011319	48	4	
Vinorelbine	IRS-mediated signalling	0.000583	0.011844	86	5	
Vinorelbine	SHC1 events in EGFR signaling	0.000669	0.013264	22	3	
Vinorelbine	SHC-mediated signalling	0.000669	0.013264	22	3	
Vinorelbine	G-protein beta:gamma signalling	0.000694	0.013600	51	4	
Vinorelbine	IRS-related events triggered by IGF1R	0.000718	0.013900	90	5	
Vinorelbine	Growth hormone receptor signaling	0.000870	0.016635	24	3	
Vinorelbine	Activation of BH3-only proteins	0.000983	0.018369	25	3	
Vinorelbine	SHC-related events triggered by IGF1R	0.000983	0.018369	25	3	
Vinorelbine	Antigen activates B Cell Receptor (BCR) leading to generation of second messengers	0.001059	0.019578	57	4	
Vinorelbine	Signaling by Leptin	0.001105	0.020001	26	3	
Vinorelbine	CLEC7A (Dectin-1) signaling	0.001106	0.020001	99	5	
Vinorelbine	Constitutive Signaling by Aberrant PI3K in Cancer	0.001367	0.024339	61	4	
Vinorelbine	SHC1 events in ERBB4 signaling	0.001376	0.024339	28	3	
Vinorelbine	Apoptosis	0.001649	0.028564	160	6	
Vinorelbine	GRB2 events in ERBB2 signaling	0.001686	0.028564	30	3	
Vinorelbine	Hemostasis	0.001698	0.028564	497	11	
Vinorelbine	Negative regulation of the PI3K/AKT network	0.001702	0.028564	8	2	
Vinorelbine	Release of eIF4E	0.001702	0.028564	8	2	
Vinorelbine	Programmed Cell Death	0.001813	0.030122	163	6	
Vinorelbine	TGF-beta receptor signaling activates SMADs	0.002037	0.033502	32	3	
Vinorelbine	AKT phosphorylates targets in the nucleus	0.002177	0.035451	9	2	
Vinorelbine	Glutathione conjugation	0.002228	0.035926	33	3	
Vinorelbine	PI3K Cascade	0.002276	0.036351	70	4	
Vinorelbine	EPHA-mediated growth cone collapse	0.002429	0.038426	34	3	
Vinorelbine	Signaling by TGF-beta Receptor Complex	0.002524	0.039538	72	4	
Vinorelbine	C-type lectin receptors (CLRs)	0.002888	0.044825	123	5	
Vinorelbine	S6K1-mediated signalling	0.003291	0.050602	11	2	
Vinorelbine	Intrinsic Pathway for Apoptosis	0.003348	0.050997	38	3	
Vinorelbine	Chk1/Chk2(Cds1) mediated inactivation of Cyclin B:Cdk1 complex	0.003929	0.059302	12	2	
Vinorelbine	p75 NTR receptor-mediated signalling	0.004223	0.063160	83	4	
Vinorelbine	deactivation of the beta-catenin transactivating complex	0.004454	0.066013	42	3	
Vinorelbine	Downregulation of ERBB2:ERBB3 signaling	0.004620	0.066667	13	2	
Vinorelbine	mTORC1-mediated signalling	0.004620	0.066667	13	2	
Vinorelbine	Regulation of Rheb GTPase activity by AMPK	0.004620	0.066667	13	2	
Vinorelbine	TCF dependent signaling in response to WNT	0.004891	0.069957	199	6	
Vinorelbine	FCERI mediated MAPK activation	0.004996	0.070851	87	4	
Vinorelbine	Prolactin receptor signaling	0.006156	0.086551	15	2	
Vinorelbine	EPH-Ephrin signaling	0.006567	0.091554	94	4	
Vinorelbine	G2/M DNA damage checkpoint	0.006999	0.095156	16	2	
Vinorelbine	Rap1 signalling	0.006999	0.095156	16	2	
Vinorelbine	Spry regulation of FGF signaling	0.006999	0.095156	16	2	
Vinorelbine	Developmental Biology	0.007292	0.098323	517	10	

^1^ Reactome pathways for eribulin and vinorelbine at *p* < 0.05 plus q < 0.1 were obtained using the 100 gene networks (Appendix A) as query inputs. ^2^ Both spellings, ‘signaling’ and ‘signalling,’ are used in the Reactome database. Spellings used here exactly reflect those downloaded from the database to facilitate searching for specific Reactome pathways names. ^3^ Shared between eribulin and vinorelbine at *p* < 0.05 plus q < 0.1.

## Data Availability

The data presented in this study are available in this article and in associated Appendix A accessible online.

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
