# Peer review of "Systematic Analysis of Genetic and Pathway Determinants of Eribulin Sensitivity across 100 Human Cancer Cell Lines from the Cancer Cell Line Encyclopedia (CCLE)"

_cancers, 2022, doi:10.3390/cancers14184532_

Round 1

Reviewer 1 Report

This study takes a comprehensive approach to determine biomarkers that are associated with sensitivity to eribulin versus paclitaxel and vinorelbine. Sensitivity was evaluated across 100 human cancer cell lines (Cancer Cell Line Encyclopedia, CCLE). Genes unique to each drug response were determined. Whereas no specific fingerprint genes could be identified for paclitaxel, 9 drug-specific fingerprint genes were identified for eribulin and 13 for vinorelbine, with minimal overlap between each drug. Ultimately, 4-genes could be used in MVR models to predict eribulin sensitivity with high accuracy to the IC50 dose. Of interest, eribulin sensitivity was related to cell signaling pathways, speculated to be related to the site of binding to tubulin at the (+) ends, perhaps providing accessibility to cell signaling hubs. 

This was a very comprehensive study with clearly described methodology and stringent statistical analysis. The results will be of high interest to the cancer research community.

A minor criticism is that Figure 4, a key figure, is far too small to be interpreted at the scale incorporated into the text. Increasing the size of each figure panel so that gene names are visible without magnification should be addressed prior to acceptance.

The discussion would be improved by further describing the relationships between the 4 eribulin-specific genes that predict resistance and how they may function biologically together to drive response. What do the authors predict would happen if one or more of these key genes were deleted (would eribulin sensitivity be lost)?

Author Response

Responses to Reviewer #1

The authors thank Reviewer #1 for insightful comments and suggestions, which resulted in a considerably improved manuscript. Following are our individual responses:

This was a very comprehensive study with clearly described methodology and stringent statistical analysis. The results will be of high interest to the cancer research community.

We thank Reviewer #1 for appreciating the value of our work.

A minor criticism is that Figure 4, a key figure, is far too small to be interpreted at the scale incorporated into the text. Increasing the size of each figure panel so that gene names are visible without magnification should be addressed prior to acceptance.

This same point was made by Reviewer #2. We replaced all panels of Figure 4 with high resolution images that maintain their resolution as they scale, and we increased the size of each panel. While font sizes may still look small if read only on a printed page, when read online or in a PDF using increased scale, the high resolution is maintained and all text labels are readable in all panels. We feel we struck the right balance between high resolution and minimizing additional space needed for the larger figure panels.

The discussion would be improved by further describing the relationships between the 4 eribulin-specific genes that predict resistance and how they may function biologically together to drive response. What do the authors predict would happen if one or more of these key genes were deleted (would eribulin sensitivity be lost)?

We especially thank Reviewer #1 for this excellent suggestion. We have now added 2 new paragraphs at the end of the Discussion to discuss the known roles of the 4 and 5 UFGs comprising the eribulin and vinorelbine MVR models. We point out the caveat that our data are correlational in nature so firm mechanistic links between genes and response cannot be drawn. However, we now highlight that the known roles of these UFGs involve cellular differentiation, proliferation and signaling processes, which is consistent with presumptive roles in setting response levels. Although the reviewer only asked this question about the 4 UFG eribulin MVR model, we felt a similar analysis of the 5 UFG vinorelbine model would be appropriate, so we added that as well. Finally, because firm mechanistic connections cannot be drawn from purely correlational data, we opted not to address the reviewer’s question about the impact of deletion of one or more of these genes in the manuscript itself, since further speculating on a mechanistic speculation would mainly be an academic exercise. But for the reviewer’s eyes here, we do speculate that deletion of one or more UFGs might have some effect on response, albeit a small one since the input of any individual UFG to the weight of the whole panel would be diluted in terms of altering a cellular phenotype defined by a different drug response.

Reviewer 2 Report

In this study, the authors identified 9 and 13 drug-specific unique fingerprint genes as predictive biomarkers for eribulin and vinorelbine by performing drug sensitivity and pathway analysis across 100 human cancer cell lines. In addition, the authors showed that the 4-gene and 5-gene multivariate regression models for eribulin and vinorelbine predicted mean IC50s for the most and least sensitive cell line quartiles accurately.

Below are some questions that the authors need to address:

1.    To show the clinical significance of the drug-specific predictive biomarkers, the drug-specific unique fingerprint genes (UFGs), especially the 4-gene eribulin MVR model and the 5-gene vinorelbine MVR model should be validated by clinical data.

2.    Due to the small number of UFGs, the accuracy of network propagation and Reactome pathway analyses should be validated through other analyses, for example, the gene set enrichment analysis.

3.    The pathway labels in Fig. 4E-J are too small to read.

Author Response

Responses to Reviewer #2

The authors thank Reviewer #2 for insightful comments and suggestions, which resulted in a considerably improved manuscript. Following are our individual responses:

1. To show the clinical significance of the drug-specific predictive biomarkers, the drug-specific unique fingerprint genes (UFGs), especially the 4-gene eribulin MVR model and the 5-gene vinorelbine MVR model should be validated by clinical data.

We especially thank Reviewer #2 for making this very important point. We have addressed this point at the end of the Conclusion section, which now ends as follows:
“Our results indicate that further investigation of the genes, pathways and MVR panels identified here is warranted, with further validation in both preclinical tumor models and in the clinical setting being of highest priority. Ultimately, our hope is that the current results together with such future validation work will lead to development of drug-specific, gene expression-based predictive biomarker panels for eribulin and vinorelbine that support improved therapeutic decision-making in the clinical setting.”

2. Due to the small number of UFGs, the accuracy of network propagation and Reactome pathway analyses should be validated through other analyses, for example, the gene set enrichment analysis.

We thank the reviewer for this comment, which prompted us to make the important clarification throughout the manuscript that our Reactome analyses focused on UFGs and UFG interactomes defined through network propagation, rather than broad pathway signatures associated with the drugs. Indeed, gene set enrichment analysis is inherent in the creation of UFG interactomes via network propagation. We have now gone through the manuscript and clarified (see yellow highlights) that identified Reactome pathways are associated with UFG interactomes, as opposed to broad drug-specific pathway signatures.

Also on this topic we address the following point here for Reviewer #2 but not in the text. In our view, the overall importance of the UFG interactome and Reactome analyses is secondary to identifying UFGs and creating UFG-based predictive MVR biomarker panels. The UFG interactome and Reactome analyses serve mainly to illustrate that cellular signaling pathways rather than microtubule-based mitotic control are the main cellular processes associated with eribulin and vinorelbine response. Since we feel that readers will readily understand the relative importance of identifying UFGs and creating UFG-based biomarker panels versus the more illustrative interactome and Reactome pathway analyses, we opted not to take up additional text space that would be needed to make this point which the readers most likely already understand.

3. The pathway labels in Fig. 4E-J are too small to read.

This same point was made by Reviewer #1. We replaced all panels of Figure 4 with high resolution images that maintain their resolution as they scale, and we increased the size of each panel. While font sizes may still look small if read only on a printed page, when read online or in a PDF using increased scale, the high resolution is maintained and all text labels are readable in all panels. We feel we struck the right balance between high resolution and minimizing additional space needed for the larger figure panels.

Round 2

Reviewer 2 Report

The authors have responded well to my comments.